# CDK1 Promotes Epithelial–Mesenchymal Transition and Migration of Head and Neck Squamous Carcinoma Cells by Repressing ∆Np63α-Mediated Transcriptional Regulation

**DOI:** 10.3390/ijms23137385

**Published:** 2022-07-02

**Authors:** Huimin Chen, Ke Hu, Ying Xie, Yucheng Qi, Wenjuan Li, Yaohui He, Shijie Fan, Wen Liu, Chenghua Li

**Affiliations:** 1Center of Growth, Metabolism and Aging, Key Laboratory of Biological Resources and Ecological Environment of Ministry of Education, College of Life Sciences, Sichuan University, Chengdu 610065, China; chm971014@163.com (H.C.); hk17882086096@163.com (K.H.); xy970923@stu.scu.edu.cn (Y.X.); qiyc4work@outlook.com (Y.Q.); fanshijie20080508@163.com (S.F.); 2State Key Laboratory of Cellular Stress Biology, School of Pharmaceutical Sciences, Xiamen University, Xiamen 361102, China; liwenjuan1993@163.com (W.L.); yhhe@xmu.edu.cn (Y.H.); w2liu@xmu.edu.cn (W.L.)

**Keywords:** ∆Np63α, CDK1, cell migration, epithelial–mesenchymal transition (EMT), head and neck squamous cell carcinoma (HNSCC), kinase, phosphorylation, transcriptional regulation

## Abstract

∆Np63α is a key transcription factor overexpressed in types of squamous cell carcinomas (SCCs), which represses epithelial–mesenchymal transition (EMT) and cell migration. In this study, we found that CDK1 phosphorylates ∆Np63α at the T123 site, impairing its affinity to the target promoters of its downstream genes and its regulation of them in turn. Database analysis revealed that CDK1 is overexpressed in head and neck squamous cell carcinomas (HNSCCs), especially the metastatic HNSCCs, and is negatively correlated with overall survival. We further found that CDK1 promotes the EMT and migration of HNSCC cells by inhibiting ∆Np63α. Altogether, our study identified CDK1 as a novel regulator of ΔNp63α, which can modulate EMT and cell migration in HNSCCs. Our findings will help to elucidate the migration mechanism of HNSCC cells.

## 1. Introduction

Head and neck squamous cell carcinomas (HNSCCs) are a group of cancers derived from the mucosal epithelium in the oral cavity, pharynx and larynx. HNSCC is the sixth common cancer worldwide, with about 1 million new cases and a half million deaths per year [1]. Owing to the tendency of regional lymph node metastasis and local recurrence, HNSCC has a low five-year survival rate [2,3,4]. The epithelial–mesenchymal transition (EMT) is a process by which epithelium-derived cancer cells increase their migration ability during the initiation of metastasis [5]. Although it is highly controversial as to whether the EMT is indispensable for cancer metastasis [6,7,8,9], there is mounting evidence of EMT in HNSCCs: by means of multi-omic analyses, EMT or partial EMT, it was found in several subtypes of HNSCC, with alterations of multiple EMT-related molecules including E-cadherin, N-cadherin, vimentin and ITGA6 [10,11,12,13,14,15]. As malignancies arising from epithelia, HNSCC cells generally express high basal levels of epithelial markers such as E-cadherin and K14 [16,17]. TGF-β and EGF, as well as the downstream ERK and AKT pathways, may be involved in activation of the EMT in HNSCCs [1,18,19]. The EMT is documented to endow HNSCC cells with properties of cancer stem cells (CSCs), including the enrichment of the CD44^+^/CD24^−^ population of cancer cells and an increased expression of CSC-related genes [20,21]. ΔNp63α, the major isoform encoded by the *p63* gene, is overexpressed in squamous cell carcinomas (SCCs) derived from multiple organs or tissues due to gene amplification [22,23]. The relationship between p63 expression and the prognosis of SCC patients is controversial: in some cases, a higher p63 level is associated with a worse prognosis [24,25,26,27]; on the contrary, there is also evidence that impaired p63 expression characterizes the biological aggressiveness of high-grade invasive carcinomas [28,29,30]. This discrepancy is likely due to the diversity of p63 isoforms, as well as their opposite roles in different SCC subtypes or in different stages [20]. As a member of the p53 family, ΔNp63α can either activate or inhibit a batch of downstream genes [22,31,32,33,34]. Recent studies have shown that ΔNp63α inhibits the EMT [22]. Mechanistically, ΔNp63α regulates genes involved in cell adhesion, motility and migration, such as integrins α6, β1 and β4 (ITGA6/B1/B4), and the matrix protein Laminin-γ2, N-cadherin, L1 cell adhesion molecule (L1CAM), bullous pemphigoid antigen 1 (BPAG1), periostin and Wnt-5a [22,35,36,37]. ΔNp63α is modulated via diverse post-translational modifications, such as phosphorylation, isomerization and ubiquitination, in which a range of enzymes are involved [38].

CDK1 (cyclin-dependent kinase 1) is an important member of the CDK family, which phosphorylates serine/threonine residues of proteins [39,40,41]. In a mixture with A- or B-type cyclins, CDK1 catalyzes the phosphorylation of more than 70 substrates, mainly regulating G1/S phase and G2/M phase transitions in the eukaryotic cell cycle [42,43,44]. In the past decade, a large body of literature has reported that CDK1 is highly expressed in hepatocellular carcinoma, lung cancer, breast cancer, thyroid cancer, colorectal cancer, prostate cancer and other malignant tumors [45,46,47,48,49]. In these scenarios, CDK1 plays key roles in the progression of cancer and survival of cancer cells [40,41,42]. Therefore, CDK1 has become one of the prospective targets for cancer therapy [42,50,51]. According to data from mouse models and patients, inhibition of CDK1 may be an effective strategy to suppress the progression of a range of malignancies [42,45,52]. In addition, recent clinical studies have demonstrated the high level of CDK1 expression in HNSCC patients and its use as a diagnostic biomarker for HNSCCs [50,51]. However, the exact roles and mechanism of this kinase in the development of HNSCCs are unclear.

In this study, we reported that CDK1 mediates the phosphorylation of ∆Np63α at T123. As a consequence, the association of ∆Np63α with its target promoters and ∆Np63α-mediated transcriptional regulation is impaired. This regulation of CDK1 on ∆Np63α may lead to EMT and the migration of HNSCC cells, as well as the progression of HNSCC patients. Our findings will help to elucidate the migration mechanism of HNSCCs as well as post-translational modifications (PTMs) of ∆Np63α.

## 2. Results

### 2.1. CDK1 Binds to ∆Np63α and Phosphorylates It on Residue T123

To investigate phosphorylation modifications and interacting proteins of ∆Np63α, we conducted an immunoprecipitation (IP) experiment with cell lysate from HEK293T cells exogenously expressing ∆Np63α. With the resultant mass spectrometry (MS) analysis, we identified several ∆Np63α-interacting kinases including CDK1 (listed in Appendix A), as well as multiple phosphorylated residues in ∆Np63α (depicted in Appendix A). Among these phospho-sites, 15 are located in the DNA binding domain (DBD) or trans-inhibitory domain (TID), both of which are crucial to ∆Np63α-mediated transcriptional regulation [53]. To investigate the effects of these phospho-sites within DBD and TID on ∆Np63α-mediated transcriptional regulation, we generated a series of mutant ∆Np63α constructs, with serine (S) or threonine (T) substituted with alanine (A) to prevent the potential phosphorylation. By means of a luciferase reporter assay using these mutant constructs, with the wild-type ∆Np63α as a control, we found that T123A mutation significantly affects ∆Np63α-mediated transactivation (Appendix A). Bioinformatic analysis reveals that the fragment of ∆Np63α containing T123 fits the consensus sequence for CDK1, and this fragment is highly conservative in species ranging from Xenopus to humans (Appendix A). Therefore, we focus on the potential CDK1-mediated modification of ∆Np63α.

In this study, the physical interaction between endogenous CDK1 and exogenous ∆Np63α was verified with an independent co-immunoprecipitation (Co-IP) experiment (Figure 1A). UM1 is a cell line derived from human head and neck squamous cell carcinoma (HNSCC), which expresses a high level of ∆Np63α [54]. To further verify the interaction between the endogenous proteins, we conducted Co-IP experiments with specific antibodies to ∆Np63α and CDK1, respectively. Our results demonstrate that endogenous proteins of ∆Np63α and CDK1 form a stable complex in UM1 cells (Figure 1B,C). These results suggest that either exogenous or endogenous proteins of ∆Np63α and CDK1 physically interact in cells.

Furthermore, we co-expressed exogenous wild-type or T123A mutant ∆Np63α with CDK1 in HEK293 cells. The wild-type or mutant ∆Np63α proteins were enriched by means of IP. Then the phosphorylation levels of ∆Np63α in the IP product were detected with the phospho-Thr (P-Thr) antibody (Figure 1D). Our results show that CDK1 can significantly increase the threonine phosphorylation level of wild-type ∆Np63α but not that of T123A mutant (Figure 1D). The weak P-Thr bands of T123A mutant suggest other phosphorylated threonine residues in the ∆Np63α protein, which are likely mediated by kinases other than CDK1. This is consistent with our IP-MS analysis, which demonstrates several phosphorylated threonine residues in ∆Np63α (Appendix A). To test the modification of endogenous proteins, we knocked down CDK1 with lentivirus-based shRNA in UM1 cells, with which we conducted IP and IB analysis. The results demonstrate that CDK1 knockdown leads to a lower phosphorylation level of threonine (P-Thr) in ∆Np63α (Figure 1E). These data suggest that CDK1 phosphorylates ∆Np63α at T123 in cells.

### 2.2. CDK1 Impairs Association of ∆Np63α with Its Downstream Gene Promoters to Inhibit ∆Np63α-Mediated Transcriptional Regulation

Next, we investigated whether CDK1 affects the transcriptional activity of ∆Np63α using luciferase reporters, respectively, driven by promoters of K14 and BPAG1, both of which are downstream target genes of ∆Np63α [22]. The luciferase reporter assays reveal that overexpression of CDK1 impairs ∆Np63α-mediated luciferase expression driven by K14 or BPAG1 promoter (K14-Luc or BPAG1-Luc, Figure 2A,B). Our further data demonstrate that CDK1 suppresses ∆Np63α-mediated K14-Luc expression in a dose-dependent manner (Figure 2C). To investigate the effects of endogenous CDK1 on the transactivity of ∆Np63α, we knocked down CDK1 and performed luciferase reporter assays. The results reveal that knockdown of CDK1 induces a significant augmentation of the ∆Np63α-mediated expression of either K14-Luc or BPAG1-Luc (Figure 2D,E). On the other hand, we treated the cells with RO3306, which is a specific CDK1 inhibitor [45]. The results reveal that RO3306 enhances ∆Np63α-mediated K14-Luc expression in a dose-dependent manner (Figure 2F). These data suggest that CDK1 inhibits ∆Np63α-mediated transcriptional regulation. 

As mentioned above, T123, the residue phosphorylated by CDK1, resides in the DNA binding domain (DBD) of ∆Np63α (Appendix A). To investigate whether CDK1 affects the association between ∆Np63α and the promoters of its downstream genes, we performed an experiment of chromatin immunoprecipitation followed by quantitative polymerase chain reaction (ChIP-qPCR) with cells overexpressing Flag-tagged ∆Np63α in combination with or without CDK1. The fragments of BPAG1 and N-Cadherin promoters were previously reported as direct targets of ∆Np63α [55], which was confirmed in our experiments (Appendix A). Our ChIP-qPCR results demonstrate that promoter fragments of BPAG1 and N-Cadherin are detected in the ChIP products, while the simultaneous expression of exogenous CDK1 decreases the precipitated fragments of BPAG1 and N-Cadherin promoters (Figure 2G). Together, these results suggest that CDK1 reduces the affinity between ∆Np63α and promoters of its downstream genes and, consequently, inhibits ∆Np63α-mediated transcriptional regulation.

### 2.3. T123A Mutation in ∆Np63α Abrogates the Inhibitory Effect of CDK1 on ∆Np63α-Mediated Transcriptional Regulation

Our aforesaid data indicate that T123 is a residue relevant to ∆Np63α activity (Appendix A), which is negatively regulated by CDK1 (Figure 2). To further confirm this hypothesis, we performed luciferase reporter assays with T123A mutant ∆Np63α, comparing its transactivity to that of wild-type ∆Np63α as well as the effects of CDK1 on them. Our data demonstrate that T123A mutant ∆Np63α has a higher transactivity than the wild type has (Figure 3A–D). This is consistent with our data shown in Appendix A. On the other hand, both the overexpression and knockdown of CDK1 fail to affect the luciferase expression mediated by T123A mutant (Figure 3A–D). These data indicate that CDK1 inhibits the transactivity of ∆Np63α by phosphorylating this transcription factor at T123.

### 2.4. CDK1 Is Upregulated in Head and Neck Squamous Cell Carcinomas (HNSCCs) and Is Correlated to Malignant Progression of HNSCCs

∆Np63α plays key roles in cell migration and metastasis of head and neck squamous cell carcinomas (HNSCCs) [22]. To explore the correlation between HNSCC and the expression of CDK1, bioinformatics analysis was performed using the Kaplan–Meier plotter database. The result demonstrated that CDK1 expression is significantly higher in HNSCCs than in normal tissues; in metastatic HNSCCs, CDK1 is further upregulated (Figure 4A). The survival analysis shows that HNSCC patients with a high expression of CDK1 have lower survival rates than those with low CDK1 expression (Figure 4B). These data indicate that CDK1 plays key roles in tumorigenesis and the progression of HNSCCs, especially in metastasis of HNSCCs.

### 2.5. CDK1 Promotes Epithelial–Mesenchymal Transition (EMT) and Migration of HNSCC Cells

To investigate the roles of CDK1 in HNSCC cell migration, we generated stable cell lines overexpressing exogenous CDK1 and performed transwell assays. As shown in Figure 5A, the Flag-tagged CDK1 was exogenously expressed in UM1 and Cal27 cells, both of which are derived from HNSCCs. The results of the transwell assay show that cells stably overexpressing CDK1 possess a higher migration ability, compared with the control cells (Figure 5B,C). Our RT-qPCR results reveal that EMT markers, N-cadherin and vimentin [56], are significantly upregulated in CDK1-overexpressing cells (Figure 5D). On the other hand, we knocked down endogenous CDK1 in UM1 and Cal27 cells (Figure 5E) and performed transwell assays (Figure 5F,G). The results reveal that the depletion of CDK1 induces a significant decrease in the migration of these cells. The RT-qPCR analyses demonstrate the significant downregulation of EMT markers, N-Cadherin and vimentin, in CDK1-knockdown cells (Figure 5H). Together, these data suggest that CDK1 promotes EMT and cell migration in HNSCC cells.

### 2.6. CDK1 Impairs Inhibitions of HNSCC Cell EMT and Migration Induced by Wild-Type ∆Np63α but Not Those by Its T123A Mutant

As mentioned above, ∆Np63α can inhibit EMT and migration in HNSCC cells [22]. To investigate whether CDK1 can modulate ∆Np63α-induced inhibition of EMT and cell migration via phosphorylating ∆Np63α at T123, we overexpressed ∆Np63α or its T123A mutant with or without CDK1 in UM1 cells (Figure 6A), with which we conducted transwell assays and RT-qPCR (Figure 6B–D). The results of transwell assays reveal that overexpression of ∆Np63α induces a significant inhibition of cell migration, while the T123A mutation further exacerbates the migration inhibition; CDK1 significantly impairs cell migration inhibition induced by ∆Np63α, but not that by its T123A mutant (Figure 6B,C). The RT-qPCR analyses demonstrate that mesenchymal markers, N-cadherin and vimentin [56], are significantly downregulated in UM1 cells overexpressing ∆Np63α, and are restored by the simultaneous overexpression of CDK1 (Figure 6D). In parallel, ∆Np63α significantly induces the expression of BPAG1 and ITGA6, which are markers of epithelial characteristics and inhibit cell migration [57,58]; the simultaneous overexpression of CDK1 can significantly impair the ∆Np63α-induced upregulation of BPAG1 and ITGA6 (Figure 6D). These results suggest that CDK1 impairs the EMT and migration inhibitions induced by ∆Np63α by phosphorylating its T123 residue.

### 2.7. Knockdown of ∆Np63α Significantly Rescues Migration Inhibition Induced by CDK1 Depletion in HNSCC Cells

To confirm the involvement of ∆Np63α in the CDK1-modulated migration of HNSCC cells, we knocked down endogenous CDK1 and (or) ∆Np63α in UM1 cells (Figure 7A). By means of transwell assays using these stable cell lines, we found that the depletion of ∆Np63α induces the enhanced migration of UM1 cells and significantly rescues migration inhibition induced by knockdown of CDK1 (Figure 7B,C). On the other hand, the ablation of CDK1 can still attenuate cell migration even in the scenario where ∆Np63α is depleted to an undetectable level (Figure 7). This indicates that CDK1 promotes cell migration, at least in part, via impairing the inhibitory effect of ∆Np63α on the migration ability of HNSCC cells. 

## 3. Discussion

Exploring the cell migration mechanism of cancer cells is of great significance for cancer treatment, especially for head and neck squamous cell carcinomas (HNSCCs), which tends to metastasize to proximal lymph nodes [59,60]. ∆Np63α is the predominant isoform encoded by *p63* gene, and plays key roles in the tumorigenesis of a variety of squamous cell carcinomas (SCCs) [61,62]. Generally, ∆Np63α is upregulated in SCCs due to gene amplification. It can promote neoplasia of SCCs by facilitating cell proliferation, while it inhibits cancer invasion and metastasis at late stages of SCCs [22]. Mechanistically, it can inhibit epithelial–mesenchymal transition (EMT) and cell migration by regulating the expression of various genes related to cell adhesion and motility [35,37]. Previous data from our lab and other groups reveal that p63 proteins can be phosphorylated and modulated by several kinases [53,63,64,65,66,67,68].

According to our IP-MS analysis, we identified the T123 in ∆Np63α is one of the phospho-sites affecting transactivity (Appendix A). On the other hand, CDK1 is one of the candidate ∆Np63α-interacting proteins (Appendix A), whose consensus binding sequence fits the fragment containing T123 (Appendix A). According to our data, there are several phospho-sites affecting the transactivity of ∆Np63α in addition to T123, though the T123A mutation leads to the biggest rise in ∆Np63α transactivity (Appendix A). It remains to be investigated which kinases mediate the phosphorylation of these residues in our system. In this study, we confirmed that CDK1 directly binds to ∆Np63α and phosphorylates it at T123 (Figure 1). Our data demonstrate that CDK1 significantly inhibits ∆Np63α-mediated transcriptional regulation (Figure 2A–F). According to our result of the chromatin immunoprecipitation (ChIP) assay, CDK1 impairs the association between ∆Np63α and the promoters of its downstream genes, including the targets of both transactivation (BPAG1) and transinhibition (N-Cadherin) (Figure 2G). On the other hand, the T123A mutation in ∆Np63α, which eliminates phosphorylation mediated by CDK1 (Figure 1D,E), abrogates the inhibitory effect of CDK1 on ∆Np63α-mediated transcriptional regulation (Figure 3). These data suggest that the CDK1-mediated phosphorylation of ∆Np63α at T123 impairs the affinity between this transcription factor and its target promoters, resulting in weakened regulation of ∆Np63α on its downstream genes. 

CDK1 is a cell cycle-related kinase, which can promote the cell cycle via phosphorylating key substrates including Rb and transactivating genes that are regulated by E2F [44]. Due to its pivotal roles in cell proliferation, diverse CDK1 inhibitors have been explored, such as flavopiridol, dinaciclib, AZD5438, AT-7519 and RO3306. In clinical or preclinical trials, these inhibitors demonstrate significant inhibition on malignancies including chronic lymphocytic leukemia, mantle cell lymphoma, hepatocellular carcinoma and gastrointestinal stromal tumor [42,45,52]. On the other hand, these CDK inhibitors may cause some adverse effects, e.g., the likelihood of recurrence induced by senescent cells in palbociclib treatment [69,70]. It remains to be seen how to avoid the side effects and whether they may apply to other cancer types [42]. In our study, we did observe the pro-proliferative effect of CDK1 on HNSCC cells, which is independent of ∆Np63α (data not shown). In this paper, we reported that CDK1 promotes the migration of HNSCC cells by modulating ∆Np63α: the overexpression of CDK1 enhances cell migration (Figure 5B,C) and significantly impairs the migration inhibition induced by the overexpression of wild-type ∆Np63α, but not that by its T123A mutant (Figure 6B,C); knockdown of CDK1 induces an inhibition of cell migration (Figure 5F,G), which can be partially rescued by the simultaneous ablation of ∆Np63α (Figure 7). It has been previously reported that CDK1 promotes the migration and metastasis of cancer cells, e.g., colorectal carcinoma and breast cancer cells, by phosphorylating proteins such as EZH2 and activating pathways such as the Wnt/β-Catenin and ERK/GSK3β/SNAI axes [71,72,73]. In our study, we observed that the depletion of ∆Np63α fails to completely rescue the migration inhibition induced by CDK1 knockdown (Figure 7). This indicates that CDK1 may also promote HNSCC cell migration in ∆Np63α-independent pathways (Figure 7). Recent studies suggest that the basal subtypes of HNSCCs undergo the EMT process (at least partial EMT) during metastasis, while mesenchymal, classical and atypical subtypes metastasize independently of the EMT [13,15]. Our in vitro investigation indicates that CDK1 modulates the ∆Np63α-mediated inhibition of EMT in basal HNSCC cells, UM1 and Cal27: CDK1 overexpression upregulates (Figure 5D), while the ablation of CDK1 downregulates (Figure 5H), mesenchymal markers N-Cadherin and vimentin; ∆Np63α-induced downregulation of mesenchymal markers (N-Cadherin and vimentin), as well as upregulation of epithelial markers (BPAG1 and ITGA6), are impaired by CDK1 (Figure 6D). Our results are consistent with data from other groups, which reveal that CDK1 promotes the migration of lung cancer cells and colorectal cancer cells via different mechanisms [72,74]. Our data show that CDK1 may, at least in part, promote the EMT and migration of HNSCC cells by phosphorylating ∆Np63α and inhibiting ∆Np63α-mediated transcriptional regulation. 

The results of the database analysis reveal that the expression of CDK1 is upregulated in HNSCCs, especially in metastatic ones (Figure 4A). On the other hand, a high level of CDK1 correlates with a shorter survival of HNSCC patients (Figure 4B). In combination with the fact that ∆Np63α is a key transcription factor controlling cancer cell migration and metastasis in multiple SCC cancer types [22], our data indicate that CDK1 may facilitate metastasis of HNSCCs, by phosphorylating and inhibiting ∆Np63α. Despite a lack of direct in vivo evidence, our findings shed new light on the mechanisms of the EMT and migration of HNSCC cells. Given the differences between cultured cells and actual tumors, this CDK1-mediated regulation of ∆Np63α, as well as its effects on EMT and cell migration, will be further verified in mouse models in our future work. It is also worthwhile to test whether CDK1 inhibitors can effectively intervene in the progression of HNSCCs in animals and patients. 

## 4. Materials and Methods

### 4.1. Cell Lines and Cell Culture

The SV40-transformed human embryonic kidney cell line HEK-293T, as well as HNSCC cell line Cal27, were purchased from Procell and cultured in Dulbecco’s modified Eagle’s medium (DMEM) supplemented with 10% fetal bovine serum (FBS) (Biology industry, Shanghai, China) and 1% penicillin and streptomycin (Biosharp, Shanghai, China). The human head and neck squamous cell line UM1 was a generous gift from Prof. Anxun Wang (Department of Oral and Maxillofacial Surgery, First Affiliated Hospital, Sun Yat-Sen University, Guangzhou, China) [75,76,77,78], and cultured in DMEM/F12 medium (Hyclone, UT, USA) supplemented with 5% FBS and 1% penicillin and streptomycin. Cell culture was carried out at 37 °C, 5% CO_2_ and 95% humidity.

### 4.2. Lentivirus Packaging and Infection

Plasmids of pLvx-Flag-ΔNp63α, pLenti6-Flag-ΔNp63α, pLKO.1-shGFP, pLKO.1-shΔNp63α have been described previously [54,79,80]. We used lentiviral-mediated shRNA interference to achieve CDK1 knockdown in tongue squamous cells. The target sequences of CDK1 shRNA were as follows: shCDK1#1, 5′-ATAGTCCTGTAAAGATTCCAC-3′; shCDK1#2, 5′-AATTAGAAGACGAAGTACAGC-3′. HEK293T cells were transfected with pLvx-Flag-CDK1/pLvx-puro/pLKO.1-shGFP/pLKO.1-shCDK1 along with the lentiviral packaging plasmids psPAX2 and pMD.2G, using Entranster^TM^-H4000 (Engreen Biosystem, Beijing, China) as a transfection reagent. The viral particles were collected 48 h post transfection. Then the particles were supplemented into the medium to incubate with UM1 or Cal27 cells overnight in the presence of 10 μg/mL polybrene. Forty-eight hours post infection, the cells were selected in a medium supplemented with 5 μM puromycin (Puro) or (and) blasticidin (BSD) for 72 h.

### 4.3. Bioinformatics Analyses

The mRNA level of CDK1 in tumor versus normal tissues and the survival analysis of CDK1 in HNSCCs were determined on website http://www.kmplot.com (accessed on 18 February 2021).

### 4.4. Immunoblot (IB) Analysis

Cells were lysed in EBC250 buffer (50 mmol/L Tris-HCl, pH 8.0, 250 mmol/L NaCl, 0.5% Nonidet P-40, 0.2% phenylmethylsulphonyl fluoride, 2 μg/mL leupeptin, 2 μg/mL aprotinin, 50 mmol/L NaF and 0.5 mmol/L Na_3_VO_4_) on ice for 30 min. Protein concentration was determined using the Bio-Rad protein assay reagent (Bio-Rad). Equal amounts of total protein were separated by 12% SDS-PAGE, transferred to a PVDF membrane and hybridized to an appropriate primary antibody and HRP-conjugated secondary antibody for subsequent detection by ECL (Beyotime) as described previously [13,81]. The primary antibodies used in this study were 1:1000 anti-P-Thr (sc-5267, Santa Claus, CA, USA), anti-CDK1 (ET1607-51, HuaBio, Hangzhou, China), anti-p63 (ET1610-44, HuaBio, Hangzhou, China), anti-Myc (EM31105, HuaBio, Hangzhou, China), anti-Flag (0912-1, HuaBio, Hangzhou, China).

### 4.5. Reverse Transcription Quantitative PCR (RT-qPCR) Analysis

Total RNA was isolated by TransZol (Transgen Biotech), followed by reverse transcription using Hifair II 1st strand cDNA Synthesis SuperMix for qPCR (Yeasen). Quantitative PCR analysis was performed in a CFX96 Real-Time PCR System (Bio-Rad) using the SYBR Green qPCR Master Mix (MCE) according to the manufacturer’s instructions. The primer sequences were as follows: 

GAPDH, 5′-GACAAGAACTCCACTTCCTG-3′ and 5′-GGCAGAGATGATGACCCTTTT-3′; 

∆Np63α, 5′-GAAGCGCCCGTTTCGTCA-3′ and 5′-CATAAGTCTCACGGCCCCTC-3′; 

CDK1, 5′-AAACTACAGGTCAAGTGGTAGCC-3′ and 5′-TCCTGCATAAGCACATCCTGA-3′; 

N-cadherin, 5′-AGCCAACCTTAACTGAGGAGT-3′ and 5′-GGCAAGTTGATTGGAGGGATG-3′; 

vimentin, 5′-TGCCGTTGAAGCTGCTAACTA-3′ and 5′-CCAGAGGGAGTGAATCCAGATTA-3′;

BPAG, 5′-GATGCAGATCCGAAAACCCCT-3′ and 5′-CTCAGTGCGGTCCAGTTGTA-3′;

ITGA6, 5′-CAGTGGAGCCGTGGTTTTG-3′ and 5′-CCACCGCCACATCATAGCC-3′.

### 4.6. Immunoprecipitation (IP) 

For the co-immunoprecipitation (Co-IP) experiment, HEK293T cells were co-transfected to express Myc-tagged CDK1 or (and) Flag-tagged ΔNp63α. Cells were lysed using cell lysis buffer for Western and IP (Beyotime) with protease and phosphatase inhibitor cocktail for general use (Beyotime) on ice for 10 min. Lysates were cleared by centrifugation at 13,000 rpm for 15 min. Protein concentration was determined using the Bradford protein assay reagent (Coolaber). Then, 2 mg of total proteins from cell lysates were incubated with anti-Flag or normal mouse IgG as control at 4 °C for 8 h, and the immune complexes were precipitated with protein A + G-agarose at 4 °C for 2 h. The immunoprecipitates were washed with lysis buffer, separated by SDS-PAGE and subjected to IB as described previously [79,80,82].

### 4.7. Chromatin Immunoprecipitation (ChIP)

ChIP assays were performed with anti-Flag (cst-#14793, Cell Signaling Technology, Beverly, MA, USA) and normal IgG antibody (KK0126, Zen Bioscience, Chengdu, China), using an Agarose ChIP Kit (Beyotime) as described in the manufacturer’s instructions. Precipitated DNA was used for qPCR, using 10% total input sheared DNA as internal controls, with the following primers: N-cadherin, forward 5′-CATCCTCAAGGGTGGGAGCT-3′ and reverse 5′-ACACAGCAAACTAAGGACGC-3′; BPAG, forward 5′-GCAGAAGTCAGACTATGATTGG-3′ and reverse 5′-TACTTCTAACGGTGAAAAGTGGC-3′. Enrichments were assessed using the ΔΔCt method, normalizing qPCR results to the mock IgG [83].

### 4.8. Transwell Migration Assay

Corning Transwell plates (Transparent PET Membrane, 24-well 8.0 μm pore size, Corning, NY, USA) without Matrigel were used to detect the longitudinal migration ability of cells. A total of 0.5 × 10^5^ cells suspended in 200 μL of culture medium with 0.1% FBS were seeded in the upper chamber, and 500 μL of medium with 10% FBS was placed in the lower chamber. After an incubation at 37 °C in a humidified 5% CO_2_ incubator for 24 h (the doubling time of UM1 and Cal27 is about 48 h), a cotton swab was used to remove the cells adhering to the membrane of the upper chamber. The migrated cells on the lower side of the filter membrane surface were stained with 0.1% crystal violet, and the fields of cells were captured by Nikon eclipse Ti–U, Japan. Five fields were randomly chosen for each well to count migrated cells to assess the migration ability as described previously [54].

### 4.9. Luciferase Reporter Assay

Luciferase assays were performed as described previously [53,84]. HEK293T cells were transfected with a mixture of K14- or BPAG1-Luc and pRL-TK-Renilla plus indicated plasmids or siRNAs. Total amount of DNAs or RNAs was balanced with control vectors or scramble control RNAs. Cells were harvested at 48 h post transfection and lysed in Passive Lysis Buffer (Beyotime). Lysates were analyzed for firefly and Renilla luciferase activities using the Dual Luciferase Reagent Assay Kit (Beyotime). Luminescence was measured in a luminometer. Relative luciferase activity was determined by normalizing luciferase activity with Renilla.

### 4.10. Statistical Analysis

Statistical analyses were conducted using GraphPad Prism 8 software. Data are presented as means ± SD from at least three independent experiments performed in triplicate; Two-tailed *t*-test or one-way ANOVA test is employed to assess the significance (***, *p* < 0.001; **, *p* < 0.01; *, *p* < 0.05). The present data are from three independent experiments performed in triplicate.

## Figures and Tables

**Figure 1 ijms-23-07385-f001:**
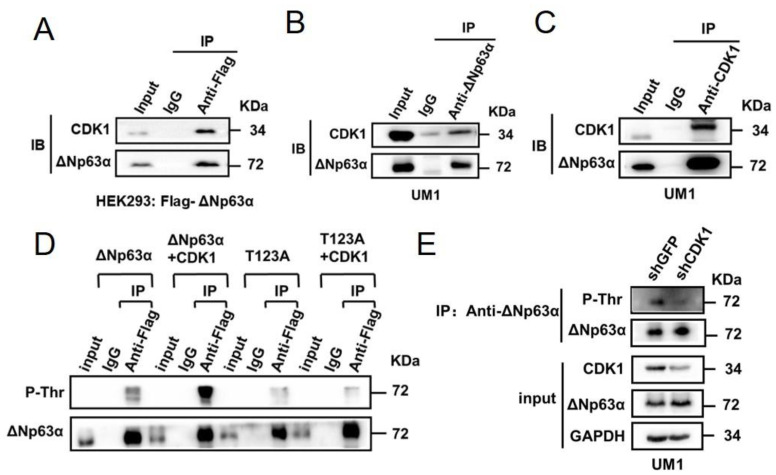
CDK1 binds to ∆Np63α and phosphorylates it at T123. (**A**) HEK293 cells transfected with Flag-∆Np63α were lysed and subjected to immunoprecipitation (IP) with anti-Flag or IgG control. The cell lysate (input) or IP products were subjected to immunoblot (IB) analysis to detect indicated proteins. (**B**,**C**) UM1 cells were lysed and subjected to IP with anti-∆Np63α or anti-CDK1, using homologous IgG as controls. The cell lysate (input) or IP products were subjected to immunoblot (IB) analysis to detect indicated proteins. (**D**) HEK293 cells transfected with Flag-∆Np63α or its T123A mutant, plus CDK1, were lysed and subjected to IP with anti-Flag or IgG control. The cell lysate (input) or IP products were subjected to IB analysis to detect indicated antigens. P-Thr, phosphorylated threonine. (**E**) UM1 cells were infected with lentivirus-based shRNA specific to CDK1 (shCDK1#1, using shGFP as a control) and selected with puromycin. The resultant stable cell lines were lysed and subjected to IP with anti-∆Np63α. Then the cell lysate (input) or IP products were subjected to immunoblot (IB) analysis to detect indicated antigens.

**Figure 2 ijms-23-07385-f002:**
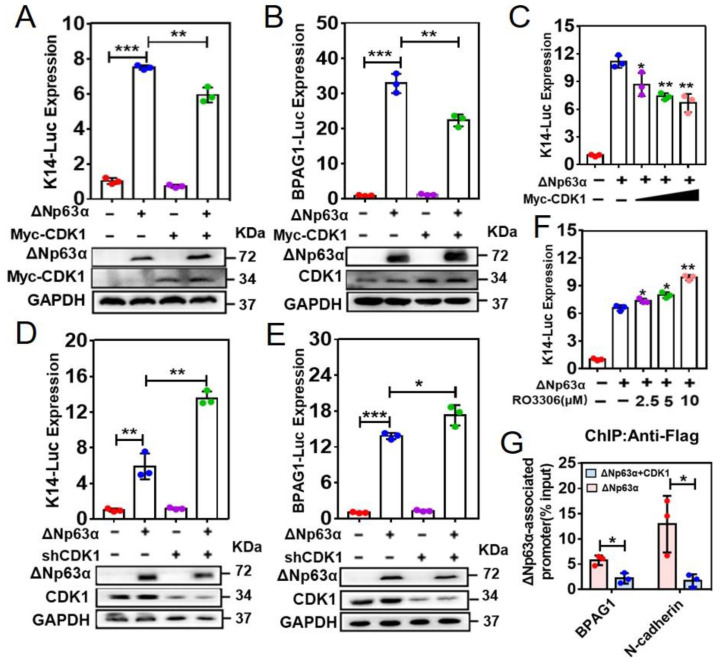
CDK1 impairs association of ∆Np63α with its downstream gene promoters to inhibit ∆Np63α-mediated transcriptional regulation. (**A**–**F**), HEK293 cells were transfected with a mixture of K14- or BPAG1-Luc and TK-Renilla, plus Flag-∆Np63α and Myc-CDK1, shCDK1#1, increasing dose of Myc-CDK1 or RO3306 (a specific CDK1 inhibitor), as indicated. Firefly and Renilla luciferase activities were measured, while IB analyses were performed to detect indicated proteins in parallel. The K14- or BPAG1-Luc expression levels were normalized to Renilla activity and presented as means ± SD (n = 3). Two-tailed *t*-test was used for comparison between two groups; ***, *p* < 0.001; **, *p* < 0.01; *, *p* < 0.05. (**G**) Chromatin immunoprecipitation (ChIP) assays were performed with UM1 cells stably overexpressing Flag-∆Np63α or Flag-∆Np63α plus CDK1, using anti-Flag. DNA samples precipitated with either anti-Flag or mock IgG, as well as equivalent input, were subjected to quantitative polymerase chain amplification (qPCR) to detect fragments of BPAG1 and N-cadherin promoters. ∆Np63α-associated promoter fragments were assessed and normalized with input and mock IgG groups (means ± SD; n = 3; *, *p* < 0.05).

**Figure 3 ijms-23-07385-f003:**
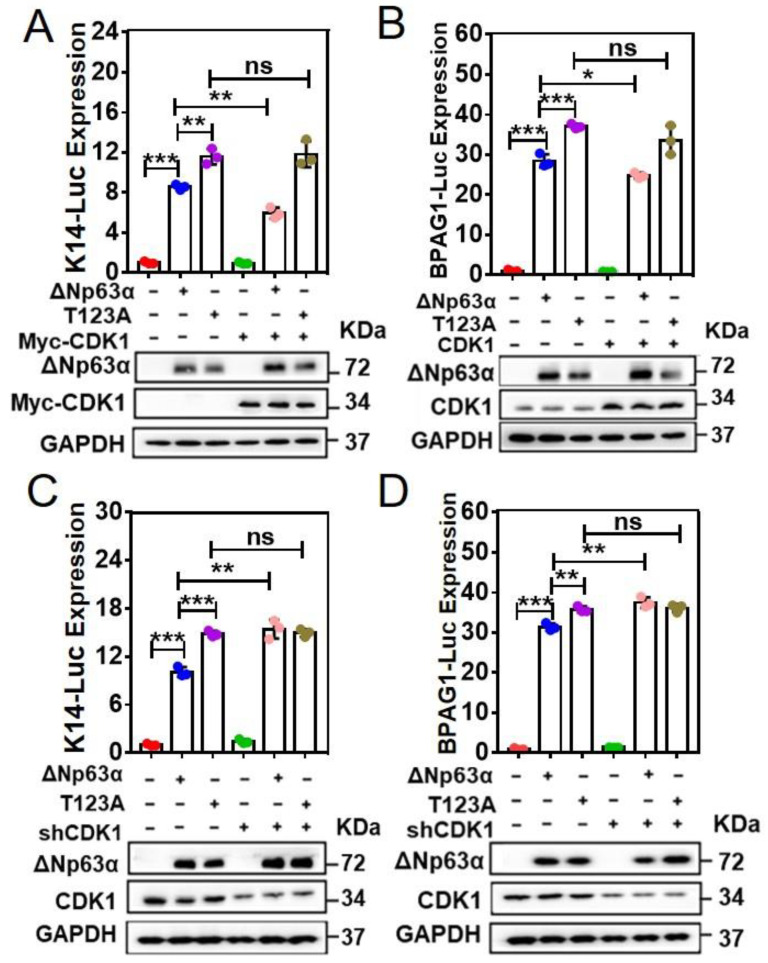
T123A mutation in ∆Np63α abrogates the inhibitory effect of CDK1 on ∆Np63α–mediated transcriptional regulation. (**A**–**D**), HEK293T cells were transfected with a mixture of K14-Luc or BPAG1-Luc and TK-Renilla, plus Flag-∆Np63α or its T123A mutant and My-CDK1 or shCDK1#1 plasmids, as indicated. IB analyses were performed to detect expression of indicated proteins in parallel. The K14-Luc or BPAG1-Luc expression levels were normalized to Renilla activity and presented as means ± SD (n = 3). Two-tailed *t*-test was used for comparison between two groups; ***, *p* < 0.001; **, *p* < 0.01; *, *p* < 0.05; ns, nonsignificant.

**Figure 4 ijms-23-07385-f004:**
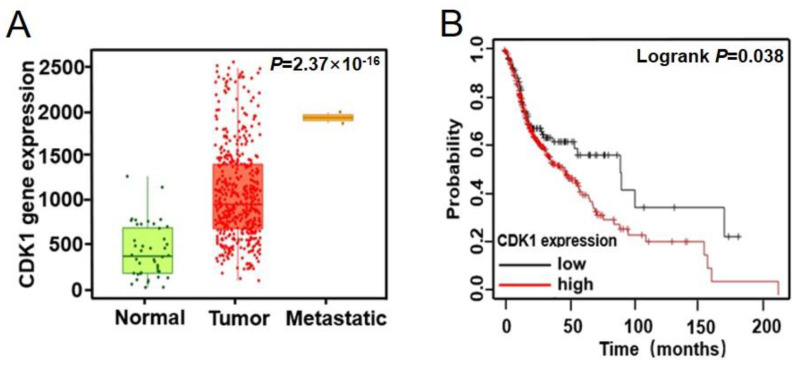
CDK1 is upregulated in head and neck squamous cell carcinomas (HNSCCs) and is correlated to malignant progression of HNSCCs. (**A**) CDK1 expression in HNSCCs and normal tissues, as well as metastatic HNSCCs, was analyzed in website http://www.kmplot.com (accessed on 18 February 2021). (**B**) The survival analysis of HNSCC patients with high or low CDK1 expression was performed in website http://www.kmplot.com (accessed on 18 February 2021).

**Figure 5 ijms-23-07385-f005:**
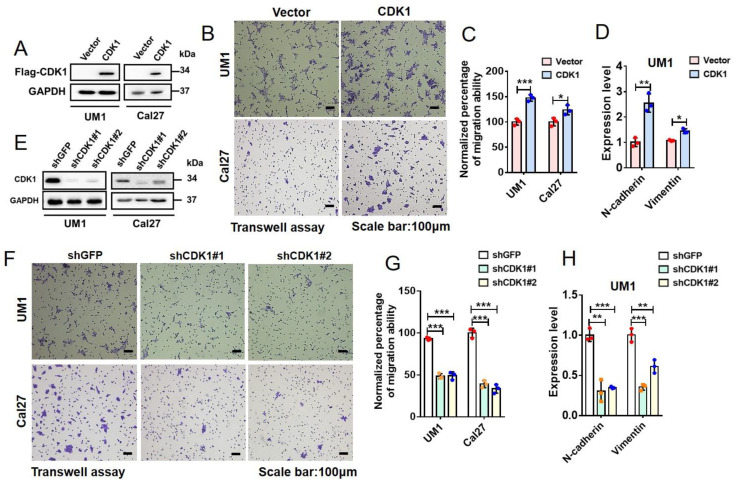
CDK1 promotes epithelial-mesenchymal transition (EMT) and migration of HNSCC cells. (**A**,**E**) CDK1 was overexpressed or knocked down in UM1 or Cal27 cells with lentiviral infection, and the resultant stable cell lines were identified by means of IB analysis. (**B**,**F**) Indicated stable cells were subjected to transwell assay. Representative crystal violet staining images of migrated cells on the underside were shown. (**C**,**G**) Quantification of effects of CDK1 overexpression or knockdown on migration ability based on transwell assays (means ± SD, n = 3). ***, *p* < 0.001; *, *p* < 0.05. (**D**,**H**) UM1 cell lines with CDK1 overexpression or knockdown were subjected to RT-qPCR analysis to measure mRNA levels of N-cadherin and vimentin (normalized with GAPDH). The qPCR data were presented as means ± SD (n = 3). ***, *p* < 0.001; **, *p* < 0.01; *, *p* < 0.05.

**Figure 6 ijms-23-07385-f006:**
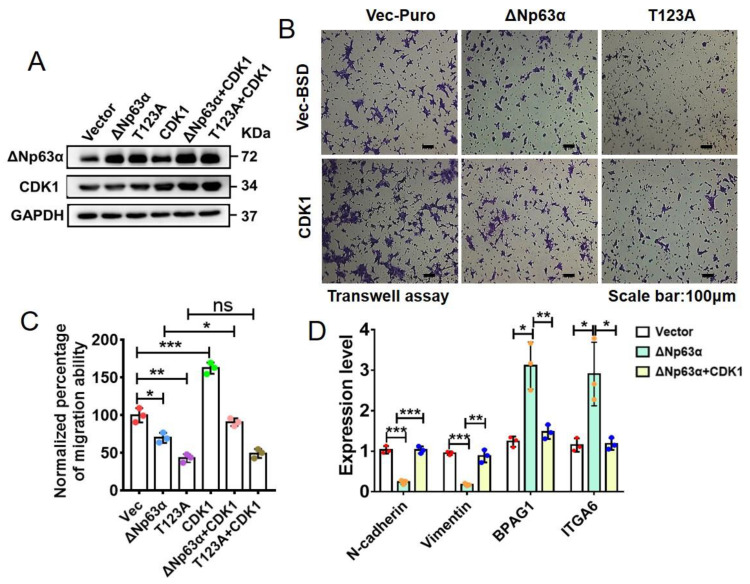
CDK1 impairs inhibitions of HNSCC cell EMT and migration induced by wild-type ∆Np63α but not those by its T123A mutant. (**A**) UM1 cells overexpressing CDK1 and (or) ∆Np63α were constructed with lentiviral infection and identified by means of IB analysis. (**B**) UM1 cells overexpressing CDK1 and (or) ∆Np63α were subjected to transwell assay. Representative crystal violet staining images of migrated cells on the underside were shown. (**C**) Quantification of effects of CDK1 and (or) ∆Np63α overexpression or ablation on migration ability based on transwell assays (means ± SD, n = 3). One-way ANOVA test was used for comparison between two groups. ***, *p* < 0.001; **, *p* < 0.01. (**D**) The abovementioned cell lines were subjected to RT-qPCR analysis to measure mRNA levels of N-cadherin, vimentin, BPAG and ITGA6 (normalized with GAPDH). The qPCR data are presented as means ± SD (n = 3). ***, *p* < 0.001; **, *p* < 0.01; *, *p* < 0.05.

**Figure 7 ijms-23-07385-f007:**
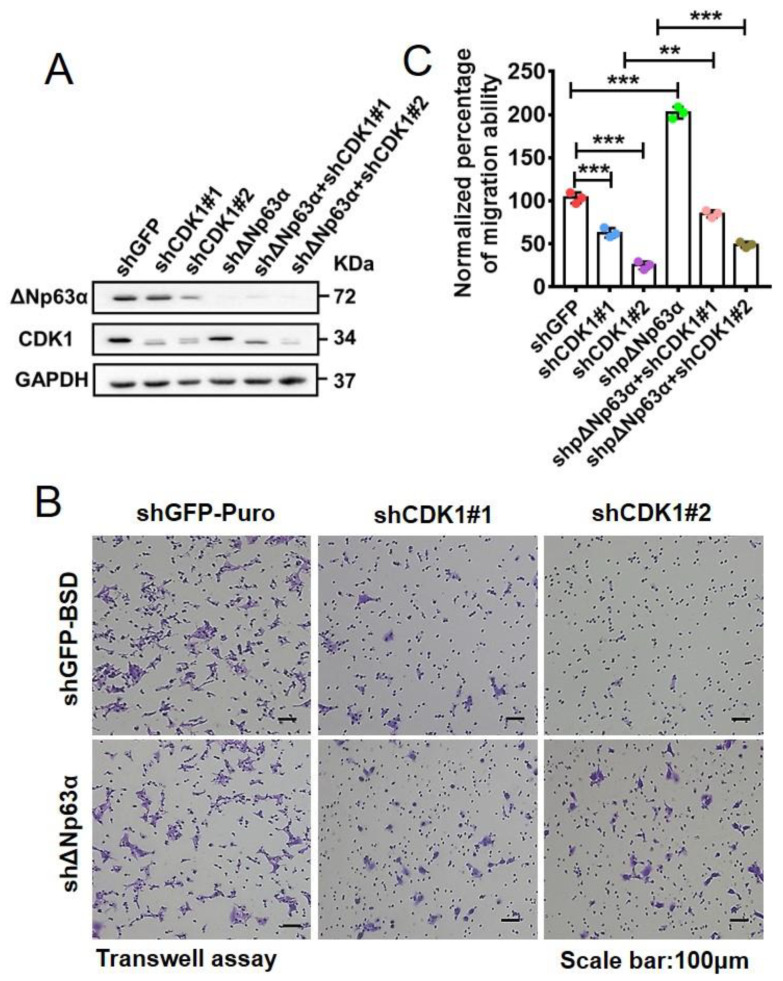
Knockdown of ∆Np63α significantly rescues inhibitions of HNSCC cell migration induced by CDK1 depletion. (**A**) CDK1- and (or) ∆Np63α-knockdown UM1 cells were constructed with lentiviral infection and identified by means of IB analysis. (**B**) UM1 cells ablated with CDK1 and (or) ∆Np63α were subjected to transwell assay. Representative crystal violet staining images of migrated cells on the underside were shown. (**C**) Quantification of effects of CDK1 and (or) ∆Np63α ablation on migration ability based on transwell assays (means ± SD, n = 3). ***, *p* < 0.001; **, *p* < 0.01.

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
