# Peer review of "CDK1 Promotes Epithelial–Mesenchymal Transition and Migration of Head and Neck Squamous Carcinoma Cells by Repressing ∆Np63α-Mediated Transcriptional Regulation"

_ijms, 2022, doi:10.3390/ijms23137385_

Round 1

Reviewer 1 Report

In my opinion the critical drawbacks of the study have been addressed and as such I would recommend to accept the publication as is.

Author Response

 Thank you very much for your assessment of our manuscript. We have revised our manuscript, with some grammars or statements improved. All changes are highlighted in a red font in the updated version.

Reviewer 2 Report

The study proposed by Chen and colleagues focuses on the interaction of CDK1 with the transcription factor ΔNp63α, known to downregulate the EMT process. The authors proposed a well-designed study where they first demonstrate that CDK1 phosphorylates ΔNp63α at the T123 site, impairing its affinity to the target promoters of its downstream genes. The authors also found overexpression of CDK1 in HNSC cancers, particularly in metastatic forms. They also identified that the inhibition of ΔNp63α conducts to an upregulation of the EMT associated with CDK1 activation. This study is very interesting since identifying the mechanisms involved in resistance and recurrences of HNSCC is highly needed to improve the prognosis of patients. By targeting CDK1, new therapeutical strategies could be proposed. Please, consider the following suggestions, which could help improve the readability of the manuscript:

 1-     I suggest to the authors improve the introduction by adding some missing information. For example, the authors said that: “there is mounting evidence of EMT in HNSCCs” (p2). Please specify some studies and develop your statement.  Is there an activation of the EMT? A higher basal level? It is linked to the CSCs proportion depending on each cell line?

The same observation is made p3 about the following sentence: “ΔNp63α, the major isoform encoded by the p63 gene, is overexpressed in squamous cell carcinomas (SCCs) derived from multiple organs or tissues, and is significantly associated with patient prognosis. “ authors should precise if it is favorable prognosis or not. Some explanations are welcome.

 2-     Concerning transwell assays, the doubling time of the cell lines used should be mentioned since usually, transwells are fixed and stained before doubling time to measure the real proportion of migrated cells, not biased by the proliferative cells. Plus, the authors should explain how many fields were counted?  Finally, transwell assay pictures should be given at higher magnificence since with the pictures presented, it is difficult to quantify isolated cells, and it looks like the membranes were not clearly and properly washed and fixed.

 3-     In Methods, the authors say that “All experiments were reproduced at least 3 times from start to finish”. The formulation is clumsy and the sentence should be rephrased.

 4-     As a general comment, there are lots of typo errors, as well as some editions of the English language used: “at ice instead of on ice”; “in the website instead of on website”….

 5-      The colors used in Supplementary figure 2 should be changed since it is difficult to distinguish green from blue.

Author Response

Attached please find the point-by-point response.

This manuscript is a resubmission of an earlier submission. The following is a list of the peer review reports and author responses from that submission.

Round 1

Reviewer 1 Report

Mechanistic study with cell transfections on CDK1 in HNSCC.

Please allow me several comments:

concerning the figures:

distribution of fig 5 not logic

fig 2 and 3: y-axis different and too big

fig 2: legend overlap

esp. fig 6: may not be an apropriate statistical test since multiple variates not only 2. overall in the MS only t-test is being used. How the results look with non-parametric tests?

Maybe other biological test needs to done to proof the mechanistical study?

What may be the next steps to go? Is there a potential clinical application? CDK1- inhibitors entering clinical trials?

problem with citations: double numbering 

Author Response

Many thanks for your pertinent comments and constructive suggestion. Attached please find our response. The changes are in red font in both this letter and the marked version of the revised manuscript.

Reviewer 2 Report

Studies examining mechanisms of tumor cell migration/metastasis are an important area of investigation however it is important that the studies focus on those processes in vivo.  This is a fully in vitro study using cultured cells that have been treated with a virus to effect the transformation.  The study is done solely in cultured cells and thus any reference to behaviors in human tumors is not appropriate as no tumors are involved.  there are no studies involved of actual tumor cell metastasis or tumor cell migration but rather only in vitro models of these processes that have many deficiencies when compared to actual tumor behavior.  the in vitro studies of CDK1 are similar to those completed by many other investigators and while the results are interesting they are no directly relevant to the stated topic to be investigated.  The in vitro findings are not verified in any tumor model and thus not directly related to the stated objectives of this research project.  Since many of the in vitro chemical analyses repeat findings completed by many other investigators there is a lack of originality in the manuscript.  The only findings related to the stated tumor cell migration focus of the study is the transwell plate analysis, which is a completely in vitro mechanism that has many limitations in comparisons to actual tumor cell invasion and migration.  The statement, "Exploring the cell migration mechanism of cancer cells is of great significant for cancer treatment" is a important statement but completely irrelevant to the study that is reported, which is not related to actual tumors, does not examine tumor cell migration and is a completely in vitro model with limited relevance to human tumors.

Author Response

We appreciate your professional comments on our manuscript. Attached please find our response. 
